# Relative Telomere Length Change in Colorectal Carcinoma and its Association with Tumor Characteristics, Gene Expression and Microsatellite Instability

**DOI:** 10.3390/cancers14092250

**Published:** 2022-04-30

**Authors:** Muhammad G. Kibriya, Maruf Raza, Mohammed Kamal, Zahidul Haq, Rupash Paul, Andrew Mareczko, Brandon L. Pierce, Habibul Ahsan, Farzana Jasmine

**Affiliations:** 1Institute for Population and Precision Health, Department of Public Health Sciences, Biological Sciences Division, The University of Chicago, Chicago, IL 60637, USA; mareczka@bsd.uchicago.edu (A.M.); bpierce@health.bsd.uchicago.edu (B.L.P.); hahsan@health.bsd.uchicago.edu (H.A.); farzana@uchicago.edu (F.J.); 2Department of Pathology, Jahurul Islam Medical College, Kishoregonj 2336, Bangladesh; drmarufraza@gmail.com; 3Department of Pathology, The Laboratory, Dhaka 1205, Bangladesh; kamal.bsmmu@gmail.com; 4Department of Surgery, Bangabandhu Sheikh Mujib Medical University, Dhaka 1000, Bangladesh; zhaq92@gmail.com; 5Department of Pathology, Cox’s Bazar Medical College, Cox’s Bazar 4700, Bangladesh; rupash_pal@yahoo.com

**Keywords:** telomere, colorectal cancer, gene expression, methylation, microsatellite instability

## Abstract

**Simple Summary:**

Telomeres are specialized repeat DNA sequences at the ends of each chromosome. The primary function of telomeres is to prevent genomic instability. Telomeres shorten with every cell division. The role of relative telomere length (RTL) change in colorectal cancer (CRC) is not fully understood. We studied RTL in CRC tissue and adjacent normal tissue from the same patients (*n* = 165). We found significant shortening of RTL in cancer tissue, irrespective of age group, gender, tumor pathology, location and microsatellite instability (MSI) status. However, shortening was more pronounced in low-grade tumors and in the presence of MSI. Gene expression data suggested an association of telomere shortening with rapid cell division (upregulation of DNA replication and cyclin-dependent kinase genes), as well as downregulation of apoptosis genes. CRC tissue showed upregulation of some telomere maintenance genes.

**Abstract:**

We compared tumor and adjacent normal tissue samples from 165 colorectal carcinoma (CRC) patients to study change in relative telomere length (RTL) and its association with different histological and molecular features. To measure RTL, we used a Luminex-based assay. We observed shorter RTL in the CRC tissue compared to paired normal tissue (RTL 0.722 ± SD 0.277 vs. 0.809 ± SD 0.242, *p* = 0.00012). This magnitude of RTL shortening (by ~0.08) in tumor tissue is equivalent to RTL shortening seen in human leukocytes over 10 years of aging measured by the same assay. RTL was shorter in cancer tissue, irrespective of age group, gender, tumor pathology, location and microsatellite instability (MSI) status. RTL shortening was more prominent in low-grade CRC and in the presence of microsatellite instability (MSI). In a subset of patients, we also examined differential gene expression of (a) telomere-related genes, (b) genes in selected cancer-related pathways and (c) genes at the genome-wide level in CRC tissues to determine the association between gene expression and RTL changes. RTL shortening in CRC was associated with (a) upregulation of DNA replication genes, cyclin dependent-kinase genes (anti-tumor suppressor) and (b) downregulation of “caspase executor”, reducing apoptosis.

## 1. Introduction

Telomeres are the terminal DNA–protein complex of chromosomes composed of repeat sequences of hexamers (TTAGGG) [1]. Proteins associated with telomeres, such as the shelterin complex, protect chromosomal ends from double-strand DNA breaks [2]. The primary function of telomeres is to prevent genomic instability [3]. Telomeres shorten with every cell division, ultimately limiting replication and potentially triggering apoptotic cell death. Telomere shortening correlates with age. The role of telomere dysfunction in colorectal carcinoma (CRC) is not fully understood. In case–control studies, it was found that relative telomere length (RTL) was longer in the colon tissue of individuals with adenoma compared to that of healthy individuals [4,5], suggesting longer RTL as a risk factor for adenoma and/or CRC. However, most studies comparing tumor tissue to paired adjacent normal tissue reveal shorter telomeres in CRC [6,7,8,9], although in some studies, no shortening observed [10].

Telomerase is the enzyme required for telomere maintenance [11] and is controlled by the telomerase reverse transcriptase (*TERT*) gene. *TERT* is downregulated in the majority of human tissues, but the telomerase RNA component (*TERC*) template is constitutively expressed [12]. Telomerase activity reappears in immortalized cell lines and in about 85% of human tumors, which has led to studies on the usefulness of telomerase for cancer diagnostics [13]. The exact mechanism of telomere length maintenance in tumor tissue is still undetermined but may be attributed to the genes related to telomerase activity, shelterin–telosome protein complex, histone binding, alternative telomere lengthening mechanism (ATL) or non-canonical telomere maintenance [14].

In this study, we examined the change in RTL in CRC tissue compared to corresponding normal colon tissue and its association with different histological and molecular changes of CRC. We also looked at differential gene expression of (a) the telomere-related genes directly or indirectly related to telomere function, (b) genes involved in selected cancer-related pathways and (c) genes at genome-wide level in CRC tissues to determine the association between gene expression and changes in RTL in paired (tumor–normal) tissues in CRC.

## 2. Materials and Methods

For this study, we included 330 paired (tumor and adjacent normal) samples from 165 CRC patients (m = 96, f = 69). Among them, 33 had right-sided CRC (cecum, 8; ascending colon, 17; hepatic flexure, 4; transverse colon, 4), and 132 had left-sided CRC (descending colon, 8; sigmoid colon, 15; rectosigmoid junction, 7; rectum, 102). A total 67 of the patients (39 male and 28 female) were aged ≤ 40 years. Fresh frozen samples were collected from 165 CRC patients from the department of Pathology, Bangabandhu Sheikh Mujib Medical University (BSMMU), Dhaka, Bangladesh, at different times spanning between December 2009 and May 2016. The patients were at different stages of CRC (Stage-1: 36; Stage-2: 42; Stage-3: 87). From each patient, the specimens were collected from the surgically resected tumor and the surrounding unaffected part of the colon about 5–10 cm away from the tumor mass. A surgical pathology fellow collected all samples from the operating room immediately after the surgical resection. Histopathology examination was performed on H&E-stained slides in routinely processed paraffin-impregnated tissue blocks. The slides were examined independently by two pathologists, and there was concordance in all 165 cases. For staging and grading of the CRC, the World Health Organization Classification of tumors was followed [15]. From each individual, we obtained a pair of tumor and normal tissues, which were frozen immediately and shipped on dry ice to the molecular genomics lab at the University of Chicago for subsequent DNA extraction and molecular assay.

For each patient, we also abstracted key demographic and clinical data and tumor characteristics from hospital medical records. Written informed consent was obtained from all participants. The research protocol was approved by the “Ethical Review Committee, Bangabandhu Sheikh Mujib Medical University”, Dhaka, Bangladesh (BSMMU/2010/10096), and by the “Biological Sciences Division, University of Chicago Hospital Institutional Review Board”, Chicago, IL, USA (10-264-E).

### 2.1. DNA and RNA Extraction and Quality Control

DNA was extracted from fresh frozen tissue using a Puregene Core kit (Qiagen, Maryland, USA). An electropherogram from Agilent BioAnalyzer with Agilent DNA 12000 chips showed the fragment size to be >10,000 bp. RNA was extracted from RNAlater preserved colonic tissue using a Ribopure tissue kit (Ambion, Austin, TX, USA, Cat# AM1924). 

### 2.2. Relative Telomere Length (RTL) Measurement

For RTL measurement, we used a Luminex-based assay using QuantiGene Plex chemistry (Invitrogen, Fisher Scientific). The details of the assay were described previously [16,17]. Briefly, the assay requires ~50 ng of DNA, which is hybridized to sequence-specific probes for the telomere repeat sequence (TEL) and reference gene sequence (*ALK)*. The TEL and *ALK* gene signals are amplified using branched DNA technology and detected using Luminex technology. We used custom-designed probes to measure abundance of the telomere repeat sequence. The 24-mer probe was targeted against 4 repeats—“TTAGGGTTAGGGTTAGGGTTAGGG”. As a reference single gene, we used *ALK*, which showed a very stable copy number (CN = 2) in all the DNA samples detected by oligonucleotide-based microarray SNP chips from our previous study. The result is a ratio; hence, there is no unit. The assay precision was good to excellent, with an intraclass correlation coefficient (ICC) of 0.91 (95% CI 0.86–0.94) [17]. The RTL assay failed in 17 samples (9 CRC and 8 normal tissue) out of the total 330 samples tested. This failure rate (5.1%) was similar to that reported in a previous large-scale study using the same Luminex-based RTL measurement assay [18].

Details of the assay and statistical analysis are presented in the Appendix A.

## 3. Results

The different patient characteristics (shown in Table 1) suggested that the CRC was similar in male and female patients in terms of age group (age ≤ 40 years, vs. >40 years), location of the tumor, pathological staging, grade, *KRAS* mutation status and *BRAF* mutation status. However, MSI was slightly more frequent among the male participants (m = 30.2% vs. f = 17.4%, *p* = 0.06, Pearson chi-square test). In this study from Bangladesh (a southeast Asian country), in both male and female patients, in ~40% of the CRC cases were diagnosed at or below the age of 40 years. In our study, CRC in the ≤40 yr age group patients was more frequently associated with advanced stage at the time of diagnosis (65.7% Stage-3 in age ≤ 40 years compared to 43.9% Stage-3 in age > 40 years group, *p* = 0.006, Pearson chi-square test).

Comparison of RTL in CRC tissue and corresponding adjacent normal colonic tissue: The change in RTL (delta) in a patient was calculated as RTL in CRC tissue minus the RTL in the corresponding normal tissue. The frequency distributions of delta RTL, as well as paired RTL values (CRC and normal), are shown in Appendix A, respectively. The RTL change in CRC tissue may also be shown as the ratio of RTL in CRC/RTL in normal tissue. However, the result was similar in respect to the biological interpretation (see Figure 1C and Appendix A, where we also adjusted for age and sex in the ANOVA model). We report both the RTL values for CRC and normal tissue to demonstrate the magnitude of difference. On average, we observed relative telomere shortening in the CRC tissue compared to paired adjacent normal colon tissue from the same patient (RTL 0.722 ± SD 0.277 vs. 0.809 ± SD 0.242, *p*= 0.00012, paired *t*-test). This magnitude of telomere shortening (by ~0.087) in tumor tissue measured by Luminex assay is equivalent to telomere shortening in human leukocytes older than ~10 year of age [17]. In our previous study, in an independent set of DNA samples from 505 individuals between 21 years and 70 years of age (m = 329, f = 176), leukocyte RTL was measured by the same assay. Linear regression analysis suggested that with an increase in age by 10 years, the leukocyte RTL decreased by 0.08 (95% CI 0.06–0.09) [17].

In stratified analysis, we looked at the differences of RTL in CRC and normal tissue (RTL in CRC minus RTL in normal tissue) according to different clinico-morphological and molecular factors (shown in Table 2). A negative mean difference indicates shortening, and a positive value indicates lengthening of telomeres in tumor tissue. “Person-to-person variation” is an important factor that explains the variation in RTL (see Figure 1). We observed statistically significant shortening of telomeres in cancer tissue, irrespective of age group, gender, location (right/left sided), pathological diagnosis, staging, grading, mutational status (*KRAS* rs112445441 and *BRAFv^600E^*) or MSI status (Table 2). RTL shortening with aging was also seen for both normal and CRC tissue (ranging from 41 to 70 years of age) in this study. Essentially parallel lines were obtained for normal and CRC tissue (see Figure 2), confirming age-related RTL shortening. The shortening was not statistically significant in patients with Stage-1 disease but was significant in Stage-2 and Stage-3 cases. Stage-2 and Stage-3 may represent more chronic cases than Stage-1 and therefore an increased number of cell divisions in the cancer tissue and more telomere shortening.

Using ANOVA models including an interaction term (Tissue × Factor), we examined if the telomere shortening in tumor tissue was significantly different in presence or absence of any particular factor. The RTL was shorter in CRC irrespective of MSI status. However in the tumor tissue, there was significantly more telomere shortening in the patients with MSI compared to the patients with MSS tumor (see Figure 3A, ANOVA interaction term *p* = 0.0059). Similarly, telomere shortening in CRC tissue was greater in patients with low-grade tumor compared to those with high-grade tumor (see Figure 3B, ANOVA interaction term *p* = 0.0043). In the same line, the telomere shortening was more pronounced in Stage-2 and Stage-3, than it was in Stage-1, but the interaction *p*-values just crossed the conventional level of significance (ANOVA interaction *p* = 0.07).

### Gene Expression of Telomere-Related Genes

In the next step, we explored the differential expression of genes related to telomere maintenance. We utilized our previous gene expression data (microarray, Illumina HT12 v4 chips) on the first 71 CRC cases and corresponding adjacent normal colonic mucosa (*n* = 70, one sample failed on microarray) from the same series of 165 cases for which we present the telomere data. Patient characteristics of the 71 CRC patients are shown in Appendix A.

Depending on the difference of RTL (RTL in CRC vs. RTL in normal tissue), we divided the patients into two groups: (a) patients with delta values < 0, termed “patients with shortening of telomere” (*n* = 52); and (b) patients with delta >= 0, termed “no shortening of Telomere” (*n* = 19). One of the most well-known genes for telomerase activity is telomerase reverse transcriptase (*TERT)*. The microarray we used had two probes in the *TERT* gene. One of the two probes showed 1.05-fold (95% CI 1.017–1.090, *p* = 0.0037) overexpression of *TERT* in CRC tissue compared to paired normal colonic mucosa (see Appendix A). This slight degree of overexpression was not different between patients with “telomere shortening” and patients with “no telomere shortening” (see Appendix A).

The complete list of telomere-related genes and paired comparisons of their expression in CRC and normal tissue is shown in Appendix A. Considering the telomere length biology, we looked for four categories of genes:Telomere maintenance genes: Among the patients with telomere shortening, gene set ANOVA analysis suggested that the “telomere maintenance” group of genes was, on average, 1.26-fold (95% CI 1.23–1.29) overexpressed in tumor tissue compared to corresponding normal colon tissue (*p* = 1.06 × 10^−68^). (Figure 4A). Even among patients without telomere shortening, these telomere maintenance genes were also overexpressed in tumor tissue compared to corresponding normal colon tissue, but by 1.17-fold (95% CI 1.13–1.1.22, *p* = 2.24 × 10^−17^) (Figure 4B). This difference in magnitude of overexpression was significantly higher in patients with “telomere shortening” (“tissue × TEL-shortening” interaction, *p* = 0.002). In other words, on average, the telomere maintenance genes were overexpressed in tumor tissue compared to paired normal colonic tissue, irrespective of the presence or absence of telomere shortening; however, the magnitude of overexpression was significantly higher if the patient had telomere shortening compared to those without telomere shortening.

2.Alternative Lengthening of Telomere genes: A list of genes is shown in Appendix A. Although statistically significant, the overall overexpression of this group of genes was only 1.03-fold (95% CI 1.01–1.05, *p* = 0.0012) in CRC tissue compared to normal tissue in patients with telomere shortening (see Appendix A). Similar analysis for the same genes in patients with no telomere shortening did not show any statistically significant differential expression (fold change =1.03 (95% CI −1.004 to 1.06, *p* = 0.089)) in CRC compared to normal tissue (Appendix A).3.Non-canonical Telomere maintenance genes: A list of genes is shown in Appendix A. These genes were, on average, 1.03-fold (95% CI 1.01–1.05, *p* = 0.0022) overexpressed in tumor tissue compared to corresponding normal colon tissue in patients with “telomere shortening” (see Appendix A). Similar analysis showed that these genes were also overexpressed to a similar extent by 1.03-fold (95% CI 1.01–1.06, *p* = 0.0052) in tumor tissue compared to corresponding normal colon tissue (see Appendix A) in patients without telomere shortening. In other words, there was no difference in differential expression of this group of genes in the presence or absence of telomere shortening (interaction, *p* = 0.76).4.Shelterin–telomere protein-related genes: These genes were not differentially expressed in CRC compared to normal colon tissue in patients with “telomere shortening” (*p* = 0.19, Appendix A). In patients with no telomere shortening, the fold change was minimal −1.03 (95% CI −1.001 to −1.06, *p* = 0.041, see Appendix A).

Telomere shortening and selected group of cancer related genes: In light of well-known biological processes involved in carcinogenesis, we examined whether telomere shortening was associated with differential expression of some selected group of genes involved in different molecular mechanisms known to be altered in cancer, such as DNA repair, apoptosis, caspase initiator, caspase executor, tumor suppressor, antitumor suppressor, etc. (see Appendix A for list). Using the “tissue x TEL-shortening” interaction term in gene set ANOVA, we first detected the functional group(s) of genes where the magnitude of differential expression was different in patients with or without telomere shortening. Then, we looked at the paired analysis stratified by telomere-shortening status. We found that there was a difference of magnitude in differential expression of some functional groups of genes, such as “caspase executor” (interaction, *p* = 1.34 × 10^−6^); “antitumor suppressor” genes, such as cyclin-dependent kinase (*CDK*) (interaction, *p* = 0.003) and “DNA repair” (interaction, *p* = 0.01) in the presence or absence of telomere shortening.

Stratified paired analysis suggested that in patients with telomere shortening, CRC tissue exhibited more overexpression of the *CDK* group of genes (1.52-fold (95% CI 1.46–1.59), Figure 5A) than the overexpression seen in patients with no RTL shortening (1.35-fold (95% CI 1.26−1.43, Figure 5B). Similarly, the DNA repair genes were also more overexpressed in patients with telomere shortening (1.09-fold (95%CI 1.08–1.11) than in patients without telomere shortening (1.05-fold (95% CI 1.03–1.08). Considering the fact that (a) overexpression of *CDK* genes (antitumor suppressor) promotes tumorigenesis and (b) DNA damage in the tumor is associated with the cellular response of upregulation in DNA repair genes, the above-mentioned results suggest an underlying association of tumorigenesis (hence, increased cell division) and telomere shortening.

On the other hand, compared to paired normal tissue, CRC tissue in patients with telomere shortening had more downregulation of caspase executor genes (−1.24-fold (95% CI −1.20 to −1.28)) than the patients without telomere shortening (−1.06-fold (95%CI −1.12 to −1.01), Figure 6A,B, respectively). This suggests an association between impaired apoptosis (tumor growth) and telomere shortening seen in CRC.

Genome-wide Differential Gene Expression at “individual gene” level: In addition to the genes known to be related to telomere length or related to known cancer pathways, we also looked at all other genes at the genome-wide level that are differentially expressed in tumor tissue compared to paired normal colonic mucosa. By including an interaction term, “tissue × telomere shortening”, in ANOVA models, we tried to identify the individual genes, the differential expression of which (CRC vs. normal tissue) differs significantly in the presence or absence of telomere shortening. The *p*-value of the interaction term in ANOVA models suggested that there was a total of 3080 probes (covering 2683 genes), the differential expression (CRC vs. normal tissue) of which differed significantly in magnitude in the presence or absence of telomere shortening. 

Gene ontology (GO) enrichment analysis of this list of genes is shown in Appendix A. The Kyoto Encyclopedia of Genes and Genomes (KEGG) pathway terms that were significantly enriched (enrichment *p*-value ≤ 0.05) within the list are shown in the Figure 7. The most significant enrichment was seen in genes related to “DNA replication”. This may suggest an association between DNA replication and telomere shortening.

Genome-wide Differential Gene Expression at “group of genes” level: In the next step, using gene set ANOVA, we tried to identify the “group of genes” (involved in different KEGG pathways), the overexpression or under-expression of which may be significantly different in magnitude in presence or absence of telomere shortening. We tested KEGG pathway terms (groups of genes). The top groups that showed difference in overall differential expression in the presence of telomere shortening are presented in Appendix A. Interestingly, the most significant group of genes was “DNA replication” (interaction *p*-value = 1.2 × 10^−11^). The other groups include “retinol metabolism”, “nucleotide excision repair”, ”base excision repair”, “mismatch repair”, etc. When we stratified the paired analysis by telomere shortening, we found that, on average, the DNA replication genes were 1.26-fold (95% CI 1.24–1.27) overexpressed in CRC compared to paired normal colonic mucosa in the presence of telomere shortening, whereas the same group of genes was 1.16-fold (95% CI 1.14–1.18) overexpressed in CRC compared to paired normal colonic mucosa in patients without telomere shortening (Figure 8A,B, respectively). This may further emphasize the association of the DNA replication process with telomere shortening. The majority of the genes in the DNA replication group (57 out of 60) were over-expressed to some degree in CRC compared to paired normal colonic mucosa. Examples of DNA replication genes are shown in Appendix A, which clearly shows that the magnitude of overexpression was greater in presence of telomere shortening.

Association of Telomere shortening, Tumor Grade and DNA replication: Our data showed that telomere shortening occurred more in low-grade tumors; we also found an association between upregulation of DNA replication genes and telomere shortening. Therefore, in the next step, using gene set ANOVA with interaction terms in the model, explored whether the overexpression of DNA replication genes was different in magnitude in low-grade and high-grade tumors. The interaction *p*-value for “tissue × grade” (*p* = 2.04 × 10^−26^) suggested that the DNA replication genes were more overexpressed in low-grade CRC than in high-grade CRC compared to paired normal samples (see Table 3). Given the fact that many of anticancer drugs work on proliferating cells, this gene expression finding may be consistent with the fact that anticancer drugs work better in low-grade than in high-grade CRC. We also included the interaction terms “tissue x telomere shortening” and “tissue × grade” in the same model as explanatory variables for the gene expression of functional groups (such as DNA replication) to explore whether the association between overexpression of DNA replication genes in CRC and telomere shortening was independent of tumor grade. The significant interaction *p*-values for both interaction terms suggest that the overexpression of DNA replication genes (presumably responsible for repeated cell division) was independently associated with telomere shortening in CRC tissue compared to paired normal colon tissue. When we looked for differential expression of telomere-related genes, we also noticed more overexpression of in low-grade CRC (see Table 3). 

Association of Telomere shortening, Tumor Stage and DNA replication: Stratified analysis by stage (see Table 3) suggested that on average, DNA replication genes were significantly overexpressed in CRC tissue compared to paired normal tissue in all stages. However, there was significantly more overexpression in Stage-1 (Table 3). When we looked for differential expression of telomere-related genes in CRC, we found that the telomere maintenance genes were significantly more overexpressed in Stage-1 CRC. In other words, our data suggested that in Stage-1, there is more overexpression of DNA replication genes (which may result in more RTL shortening), but there is also more overexpression of telomere maintenance (which may compensate for the RTL shortening). In addition, chronicity may also play a role in the slightly more telomere shortening that occurs in Stage-2 and Stage-3 compared to Stage-1, although DNA replication genes were slightly more overexpressed in Stage-1.

Association of Telomere shortening, MSI status and DNA replication: Our data showed that telomere shortening was more pronounced in MSI tumors; we also found an association between upregulation of DNA replication genes and telomere shortening. Therefore, in the next step, we explored whether the overexpression of DNA replication genes was different in magnitude in MSS and MSI tumors. We found overexpression of DNA replication genes both in MSS and MSI tumors compared to paired normal tissue without any difference in magnitude (see Table 3). In other words, the difference in RTL shortening (more in MSI tumors) could not be attributed to a difference in upregulation of DNA replication genes. When we looked for differential expression of telomere maintenance genes stratified by MSI status, we found significant overexpression of telomere maintenance genes both in MSS and MSI tumors compared to paired normal tissue without any difference in magnitude, depending on MSI status (Table 3).

Association of Telomere shortening and Expression of Immune Target genes: Previously, we showed that MSI status could be used to identify CRC patients who would potentially benefit from particular immune checkpoint inhibitor drugs [19]. We explored whether the differential expression of such target genes in CRC tissue (compared to paired normal tissue) differs by telomere shortening status. Figure 9 shows a similar pattern of differential expression in CRC tissue with or without telomere shortening. There was no significant difference (interaction *p* = 0.66). This suggests that telomere shortening may not be helpful to identify patients who may potentially benefit from immune checkpoint inhibitors.

Association of Telomere shortening and Methylation status of Telomere-related genes: In addition to looking at the gene expression pattern, we also looked at the DNA methylation pattern. We utilized our previous genome-wide methylation data (microarray, Illumina Infinium methylation 450 K chips) on the first 125 CRC cases and corresponding adjacent normal colonic mucosa (*n* = 125) from the same series of 165 cases for which we presented the telomere data. In the methylation array we used, there was a total of 236 methylation probes covering the telomere-related genes. None of the probes showed differential methylation in tumor tissue compared to corresponding normal tissue exceeding 0.14 in either direction (hypo- or hypermethylation). The magnitude of differential methylation (beta of tumor vs. beta of corresponding normal) was low regardless of *p*-value (see Figure 10). We also used stratification by telomere shortening. The differential methylation of probes in (a) telomere maintenance genes (Appendix A), (b) alternate lengthening of telomere related genes (Appendix A), (c) non-canonical telomere maintenance genes (Appendix A) and (d) shelterin complex genes (Appendix A) are shown in additional figures. In general, telomere maintenance genes were very slightly hypomethylated in tumor tissue compared to corresponding normal colonic tissue both in the presence or absence of telomere shortening. This correlated also with the gene expression finding. 

## 4. Discussion

A large-scale study including 31 cancer types suggests telomere shortening in a vast majority of cancers [20]. Cancer cells require telomere maintenance mechanisms for unlimited replicative potential. They may achieve this through *TERT* activation or alternative telomere lengthening associated with *ATRX* or *DAXX* loss. A recent large study involving 2500 matched tumor–control samples from 36 different tumor types suggests that whereas the telomere content of tumors with *ATRX* or *DAXX* mutations is increased, tumors with *TERT* modifications show a moderate decrease in telomere content [21]. It has also been proposed that telomere attrition could repress the outgrowth of cancer [22,23,24]. The role of telomere length in the etiology of CRC is not fully understood. A large study was conducted in Singapore enrolling more than 2600 subjects, among which 776 had CRC. Researchers measured telomere length from leucocytes using the qPCR method. Their findings support the notion that longer telomeres in leukocytes may be associated with a higher risk of CRC, particularly rectal cancer [25].

Fernandez-Rozadilla et al. compared telomere length of colon tissue from healthy controls with that of individuals with CRC, polyps and past or family history of polyps or CRC [4]. They found that the telomere length of normal mucosa of cases was higher than that of the controls. They found no correlation with location of tissue. Blood telomere length showed no association with case control status [4]. Peacock et al. examined the relationship between telomere length in normal colon tissue and the prevalence of colorectal adenoma, a precursor to CRC [5]. Forty cases with adenomas detected by colonoscopy and 45 controls were studied. RTL was measured using quantitative real-time PCR. RTL was significantly longer in colon tissue of individuals with adenomas compared to that of healthy individuals (*p* = 0.008) [5]. 

In some studies, investigators tested fresh frozen CRC tissue and corresponding adjacent normal colon mucosa and found a shorter telomere length in cancer tissue compared to the healthy tissue [8,9,26,27,28,29]. Suraweera et al. carried out a large study wherein they analyzed the elative telomere length of 90 adenoma, 419 CRC and their corresponding normal mucosa. Both adenoma and CRC tissue had significantly shorter RTL than the corresponding healthy mucosa [30]. They also found that female gender, proximal location and the TERT rs2736100 G allele were independently associated with longer age-adjusted RTL in normal mucosa of these patients. The age-adjusted RTL in normal and cancer tissue were independently associated with tumor stage. The authors did not find any association of tumor status of MSI, CIMP, *TP53, KRAS* and *BRAF* with RTL in normal or tumor tissue. However, they reported that near-tetraploid CRCs exhibited significantly longer RTL [30].

Rampazzo et al. studied 118 (CRC) samples (53 right colon, 30 left colon and 35 rectum) and corresponding adjacent non-cancerous tissues for telomere length, *p53* mutation and MSI [9]. Telomeres were significantly shorter in CRCs than in adjacent tissues, regardless of tumor stage and grade, site or genetic alterations (*p* = 0.0001). Telomere length in CRCs did not differ with tumor progression or *p53* status. In CRCs carrying the wild-type *p53*, telomeres were significantly shorter in tumors with MSI than in those with stable microsatellites (*p* = 0.027). Furthermore, telomere length differed according to tumor location, being longer in rectal cancers (*p* = 0.03).

O’Sullivan et al. measured telomere length by Q FISH and qPCR in colon adenoma [3]. Telomere length in larger colon adenoma lesions (>2 cm) was significantly shorter than that in normal adjacent (*p* = 0.004) or normal distant (*p* = 0.05) tissue from the same individuals. However, this association was not found in cases with colon adenoma of less than 2 cm in size.

A study was done on 42 MSS Stage-2 CRC tissue samples, where whole-exome sequencing data from paired normal colon and tumor tissue showed shorter telomere in tumor tissue [7]. The number of reads from the tumor sample was normalized to build the tumor telomere length ratio (TTLR). The functional gene set enrichment analysis showed pathways related to immune response (*TCTN3, RNF7, ATP6VOE1, SNX3* and *UBAP1*) significantly associated with TLLR. However, telomere length had no correlation with gene expression changes of most of the genes responsible for telomere maintenance, *TERT, RAP1, DKC1, TERF1, TERF2, POT1, TERF2IP* and *TPP1* [7].

Kroupa et al. [6] performed a large study with many important findings that are reproduced in our current study. They collected CRC tissue and adjacent healthy tissue from 721 individuals. RTL was measured by qPCR. RTL in tumor tissues was shorter than that in the adjacent mucosa (*p* < 0.0001). Shorter RTL was also observed in early-stage tumors compared to the advanced stages (*p* = 0.001). RTL was shorter in tumors at the proximal colon than at the distal colon (*p* < 0.0001). The authors compared paired normal and tumor tissue in patients with MSI and MSS, where patients with MSI had significantly shorter telomeres (*p* = 0.001). A similar finding was reported by Takagi et al. [31]. Telomere shortening was also shown in mucinous tumor histology (*p* < 0.0001) compared to tubular carcinoma [6]. 

One of the limitations of the current study is the lack of data on telomerase enzyme activity. The core components of telomerase are the reverse transcriptase *(TERT)* and telomerase RNA component (*TERC*), which provide the template for the synthesis of telomeric DNA. We looked at the differential expression of these genes. The role of telomerase in cancer was reviewed recently [32].Telomerase is associated with a set of accessory proteins, including dyskerin and nucleolar protein 10 (*NOP10*). We found that *DKC1* (dyskerin pseudouridine synthase 1) was overexpressed in CRC. This is consistent with results reported by Nersisyan et al. [29], who studied gene expression in paired samples (tumor vs. healthy) in Lynch syndrome, sporadic CRC with MSI and MSS [29]. The *DKC1* gene provides instructions for making a protein called dyskerin. This protein is involved in maintaining telomeric structure, which may be responsible for prolonged cell life. Alder et al. showed that mutation of the *DKC1* gene was associated with early-age morbidity due to failure of telomere maintenance [33].

Telomerase is also known to be expressed in certain non-malignant cells, although generally at low levels. Engelhardt et al. investigated telomerase activity and telomere length in premalignant, malignant, inflammatory and normal colon specimens [34]. Telomerase activity was evaluated in 130 frozen specimens from human colon cancer (*n*= 50), adjacent normal colon tissue (*n*= 50), colon polyps (*n* = 20) and colitis (*n* = 10) using a modified telomeric repeat amplification protocol assay, whereas telomere length was assessed by Southern blot. High to moderate levels of telomerase activity were detected in 90% of colorectal tumors. Weakly positive activity was detected in 10%. None of the normal tissues exhibited telomerase activity. In polyps and colitis, telomerase activity was found but was 25- to 54-fold lower than that detected in colon cancer (*p* < 0.001). Late-stage tumors demonstrated increased telomerase activity compared to early-stage tumors [34].

Katayama et al. examined telomerase activity and telomere length in gastric and colon cancer [10]. Although telomerase activity was detected in 1/12 (8%) cases of gastric polyp and in 2/9 (22%) cases of colorectal polyps, its positivity in gastric cancer and CRC was 7/10 (70%) and 21/26 (81%; *p* < 0.0003 and *p* < 0.0001, respectively) [10]. 

Gertler et al. found significant positive correlations between telomere length and *hTERT* expression in both non-cancer colorectal mucosa (r 0.54; *p* 0.001) and CRC (r 0.52; *p* 0.001) [27].

Garcia-Aranda et al. studied 91 primary CRCs and their corresponding normal tissue samples for telomere length and telomerase activity [26]. Most tumors (81.3%) displayed telomerase activity. Telomeres in CRC specimens were significantly shorter compared to the normal, adjacent specimens (*p* = 0.02). Moreover, tumors that demonstrated shortened telomeres displayed higher *TRF1* levels than tumors without telomere shortening.

Bergstrand et al. studied Hoyeraal–Hreidarsson syndrome (HHS), a multisystem disorder characterized by bone marrow failure, developmental defects and very short telomeres, which is caused by germline mutations in genes related to telomere biology [35]. *WRAP53* is an essential component of the telomerase holoenzyme complex, a ribonucleoprotein complex required for telomere synthesis. A mutation of this gene is responsible for the HHS. We found the *WDR79 (WRAP53)* gene to be overexpressed in CRC tissue compared to normal tissue, and the overexpression was more pronounced in the presence of MSI, as well as in presence of RTL shortening (Appendix A).

Fu et al. found that overexpression of the *NAT10* gene is associated with telomere shortening [36]. The shorter telomeres in the CRC tissue and *NAT10* overexpression found in our study are consistent with this finding.

Considering the fact that the RTL varies from person to person, examination of paired samples (tumor and adjacent normal tissue) from the same patient may be the source of the most robust result with respect to an association between cancer and RTL. Our current study strongly suggests that in general, there is shortening of the telomere in CRC tissue compared to adjacent normal colonic tissue. However, in a few cases, there is lengthening of telomere as well. Our data suggest that the shortening of the telomere is the consequence of rapid cell division in CRC, which is supported by its significant association with upregulation of “DNA replication” genes, upregulation of “antitumor suppressor genes (cyclin dependent kinase genes)”, upregulation of “DNA repair” genes and downregulation of “caspase executor” (reducing apoptosis). Our data also suggest that cancer cells exhibit upregulation of some “telomere maintenance” genes, irrespective of telomere shortening, although the upregulation was greater in the presence of telomere shortening. We admit that we have data on neither telomerase activity nor somatic mutation of telomere-related genes. We also do not have clinical follow-up data to test the prognostic significance of telomere shortening in CRC. Telomere shortening was more noticeable in low-grade tumors (which was associated with more upregulation of “DNA replication” genes) and in MSI tumors (which was not related to upregulation of “DNA replication” genes). The current study cannot determine the cause or effect relationship of MSI and telomere shortening. Both may represent genomic instability. A recent study by Takagi et al. also found association between MSI and telomere shortening in CRC [31].

To our knowledge, this is the first study utilizing clinical samples from a southeast Asian country. In addition, this is one of the few studies incorporating genome-wide differential gene expression, as well as differential methylation data, from a subset of the same patients to study the association with RTL change in CRC. About 40% of the patients were ≤40 years of age, who had more severe disease than the patients of >40 years. Some of our findings support the previous findings in different populations. Unlike MSI, which may be used as a surrogate marker for upregulation of some immune target genes (hence, a potential candidate for certain checkpoint inhibitors), this telomere shortening may not be a marker for the ICI group of drugs but may be a potential marker of increased cell division and hence better candidates for traditional antineoplastic drugs working on proliferating cells in general. Future studies are needed to support this assumption. 

## 5. Conclusions

In CRC, we found significant shortening of RTL in cancer tissue compared to the paired normal tissue, irrespective of age group, gender, tumor pathology, location and MSI status. RTL shortening was more frequent in low-grade CRC and in the presence of MSI. In CRC, RTL shortening is associated with (a) upregulation of DNA replication genes, cyclin dependent kinase genes (antitumor suppressor), and (b) downregulation of “caspase executor genes”, reducing apoptosis.

## Figures and Tables

**Figure 1 cancers-14-02250-f001:**
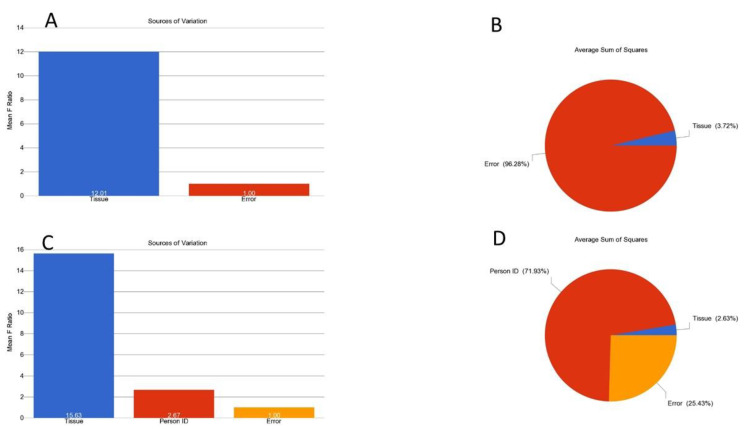
Variation of RTL in the data. The source of variations in the RTL that can be explained in the ANOVA models are shown. The upper row (**A**,**B**) shows unpaired analysis, and the lower row (**C**,**D**) shows result from paired analysis (taking the “Person ID” in the ANOVA model). The mean F-ratio (F-statistics for the factor/F-statistics for the model error) representing the significance of the factor in the ANOVA model is shown in the bar graphs (**A**,**C**). The sums of squares in the ANOVA model representing the proportion of the variation explained by the factors are shown as pie charts (**B**,**D**). When the “Person ID” is not included in the ANOVA model (unpaired analysis shown in **B**), 96% of the variation in RTL cannot be explained (error, 96.28). However, when the “Person ID” is incorporated into the ANOVA model (paired analysis shown in **D**), the error reduces to 25%, and the person-to-person variation explains 71.93% of the variation in RTL.

**Figure 2 cancers-14-02250-f002:**
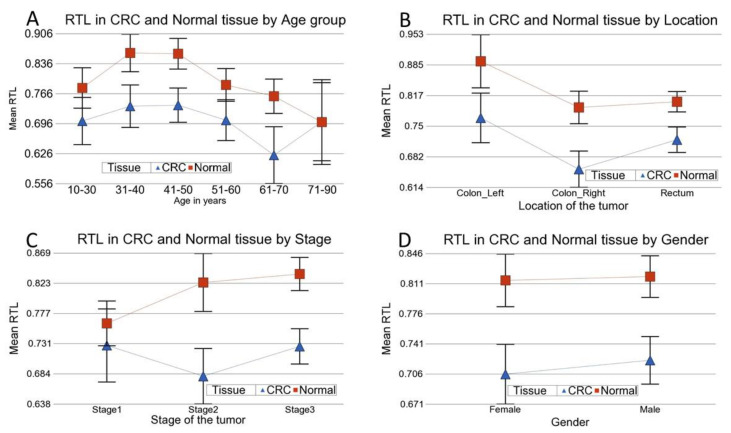
Comparison of RTL in CRC tissue (in blue) and corresponding normal tissue (in red) by age group (**A**), location of the tumor (**B**), tumor stage (**C**) and gender (**D**).

**Figure 3 cancers-14-02250-f003:**
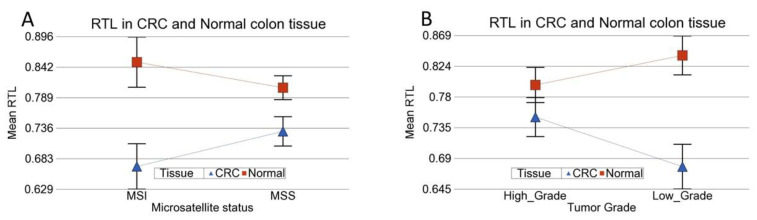
Comparison of RTL in CRC tissue (in blue) and corresponding normal tissue (in red) by MSI status (**A**) and by tumor grade (**B**). RTL shortening in CRC tissue was greater in MSI tumors, as well as in low-grade tumors.

**Figure 4 cancers-14-02250-f004:**
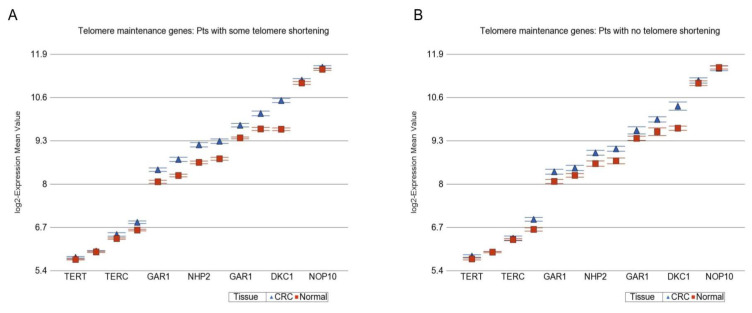
Differential gene expression of telomere maintenance genes in CRC tissue (in blue) compared to corresponding normal tissue (in red). The gene probes (multiple probes for some genes) are shown on the x-axis, and the Log2-transformed expression value is shown on the y-axis. A difference of 1 on the y-axis represents a two-fold overexpression. Data from patients with RTL shortening are shown on the left (**A**), and data from patients without RTL shortening are shown on the right (**B**). The average magnitude of overexpression was significantly higher (*p* = 0.002) if the patient had telomere shortening (1.26 fold (95% CI 1.23–1.29)) compared to those without telomere shortening (1.17 fold (95% CI 1.13–1.1.22)).

**Figure 5 cancers-14-02250-f005:**
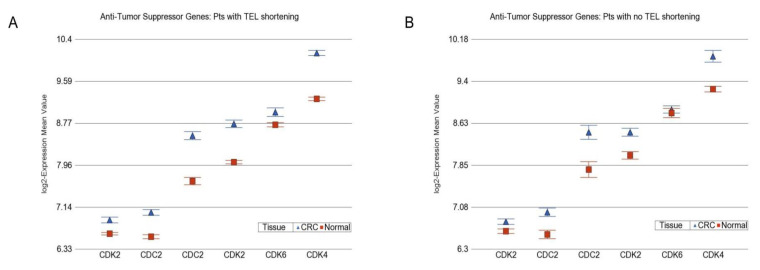
Differential gene expression of antitumor suppressor genes (cyclin D kinase) in CRC tissue (in blue) compared to corresponding normal tissue (in red). The gene probes (multiple probes for some genes) are shown on the x-axis, and the Log_2_-transformed expression value is shown on the y-axis. CDC2 is also known as CDK-1. Data from patients with RTL shortening are shown on the left (**A**), and data from patients without RTL shortening are shown on the right (**B**). The average magnitude of overexpression was significantly higher (*p* = 0.003) if the patient had telomere shortening (1.52-fold (95% CI 1.46–1.59), shown in **A**) compared to those without telomere shortening (1.35-fold (95%CI 1.26−1.43), shown in **B**).

**Figure 6 cancers-14-02250-f006:**
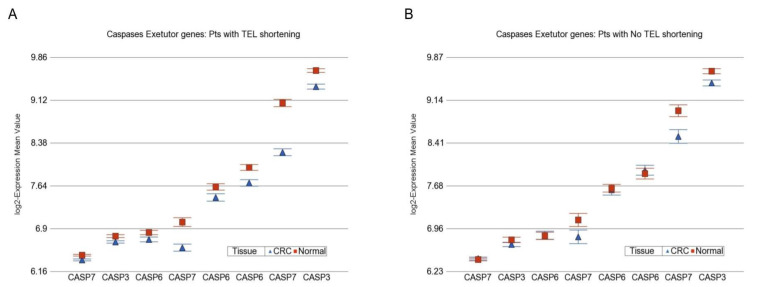
Differential gene expression of caspase executor genes in CRC tissue (in blue) compared to corresponding normal tissue (in red). The gene probes (multiple probes for some genes) are shown on the x-axis, and the Log_2_-transformed expression value is shown on the y-axis. Data from patients with RTL shortening are shown on the left (**A**), and data from patients without RTL shortening are shown on the right (**B**). The average magnitude of downregulation was significantly more (*p* = 1.34 × 10^−6^) if the patient had telomere shortening (−1.24-fold (95% CI −1.20 to −1.28), shown in **A**) compared to those without telomere shortening (−1.06-fold (95%CI −1.12 to −1.01), shown in **B**).

**Figure 7 cancers-14-02250-f007:**
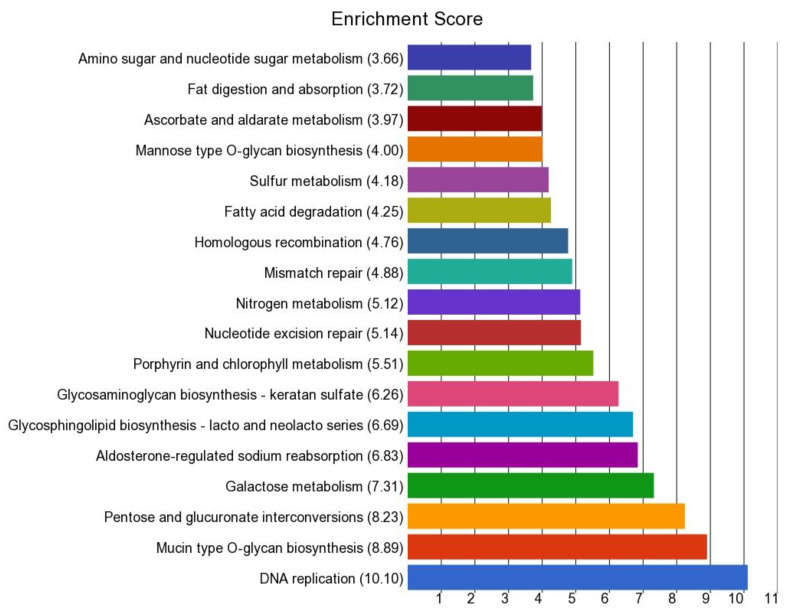
Enrichment score of the list of genes that had a significantly different magnitude of differential expression in CRC tissue than normal tissue, depending on the presence or absence of MSI.

**Figure 8 cancers-14-02250-f008:**
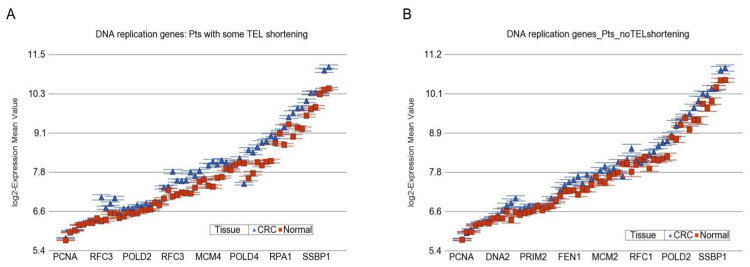
Differential expression of genes involved in “DNA replication” in CRC tissue (in blue) compared to corresponding normal tissue (in red). The gene probes (multiple probes for some genes) are shown on the x-axis, and the Log2-transformed expression value is shown on the y-axis. Data from patients with RTL shortening are shown on the left (**A**), and data from patients without RTL shortening are shown on the right (**B**).

**Figure 9 cancers-14-02250-f009:**
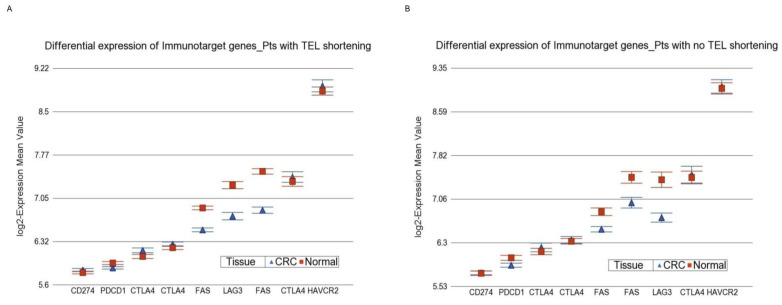
Immune checkpoint inhibitors (ICIs) are targeted to certain immunotarget genes. Differential expression of those immunotarget genes in CRC tissue (in blue) compared to corresponding normal tissue (in red) is shown. Data from patients with RTL shortening are shown on the left (**A**), and data from patients without RTL shortening are shown on the right (**B**). The magnitude of differential expression was similar in patients with or without RTL shortening.

**Figure 10 cancers-14-02250-f010:**
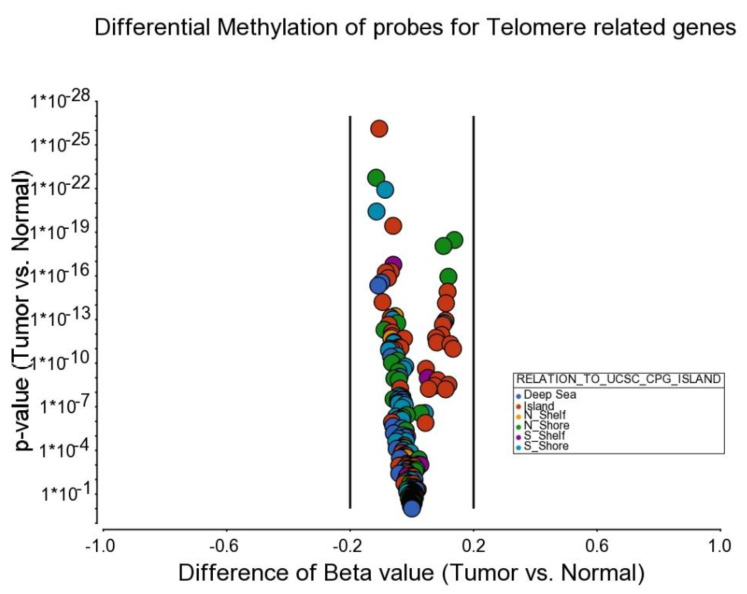
Volcano plot showing the probe-level differential methylation data of telomere-related genes in CRC tissue compared to corresponding normal tissue. Magnitude of difference (beta value of CRC minus beta value of normal tissue) is shown on the x-axis, and the *p*-value from paired t-test is shown on the y-axis. The location of the probes in relation to the CpG island are color-coded.

**Table 1 cancers-14-02250-t001:** Characteristics of the 165 CRC patients.

Characteristic	Category	Male (*n* = 96)	Female (*n* = 69)	*p*-Value
Age (years)	mean	45.6	44.86	0.773
	(SD)	13.82	13.98	
Age category	≤40 years	39	28	0.582
	>40 years	57	41	
Location	Left	75	57	0.556
	Right	21	12	
Histopathology	Adenocarcinoma	78	63	0.171
	Mucinous adenocarcinoma	17	6	
	Squamous cell carcinoma	1		
Stage	Stage-1	21	15	0.987
	Stage-2	24	18	
	Stage-3	51	36	
Grade	Low	43	35	0.528
	High	53	34	
Microsatellite	MSI	29	12	0.069
	MSS	67	57	
CEA (ng/mL)	mean	42.59	45.49	0.844
	(SD)	95.14	86.9	
*KRAS*	Wild	65	50	0.435
	Mutant	29	19	
*BRAFV^600E^*	Wild	87	64	0.327
	Mutant	6	5	

**Table 2 cancers-14-02250-t002:** RTL in CRC and paired normal tissue.

Characteristic	*n*	RTL in CRC Tissue		RTL in Normal Tissue		Delta = Tumor-Normal			*p* Value
		Mean	SD	Mean	SD	Mean	95% Lower BOUND	95% Upper Bound	
**Gender**									
Male	88	0.73	±0.27	0.81	±0.24	−0.08	−0.14	−0.02	0.009
Female	64	0.71	±0.29	0.81	±0.25	−0.09	−0.16	−0.03	0.004
**Age**									
≥40	91	0.72	±0.26	0.81	±0.23	−0.09	−0.15	−0.04	<0.001
<40	61	0.73	±0.3	0.81	±0.26	−0.08	−0.16	0.002	0.055
**KRAS**									
Mutant	45	0.67	±0.31	0.80	±0.24	−0.12	−0.21	−0.04	0.007
Wild	106	0.75	±0.26	0.82	±0.24	−0.07	−0.12	−0.02	0.007
**BRAF**									
Mutant	11	0.69	±0.21	0.82	0.18	−0.13	−0.33	0.07	0.190
Wild	139	0.73	±0.28	0.81	±0.24	−0.08	−0.13	−0.04	<0.001
**MSI Status**									
MSI	38	0.67	±0.26	0.85	±0.28	−0.18	−0.26	−0.10	<0.001
MSS	114	0.74	±0.28	0.80	±0.23	−0.06	−0.11	−0.006	0.029
**Histopathological Diagnosis**									
Adenocarcinoma	130	0.72	±0.28	0.81	±0.24	−0.09	−0.14	−0.05	<0.001
Mucinous Adenocarcinoma	21	0.75	±0.26	0.81	±0.28	−0.06	−0.20	0.09	0.444
Squamous Cell Carcinoma	1	0.97		0.84					
**Stage**									
Stage-1	34	0.74	±0.34	0.76	±0.21	−0.02	−0.14	0.09	0.692
Stage-2	37	0.70	±0.27	0.82	±0.29	−0.12	−0.19	−0.05	0.001
Stage-3	81	0.73	±0.25	0.82	±0.23	−0.10	−0.16	−0.04	0.002
**Grade**									
High-grade	78	0.75	±0.26	0.79	±0.24	−0.03	−0.08	0.03	0.313
Low-grade	74	0.69	±0.29	0.84	±0.25	−0.15	−0.21	−0.08	<0.001
**Location**									
Left	122	0.74	±0.29	0.82	±0.25	−0.08	−0.13	−0.03	0.003
Right	30	0.65	±0.24	0.28	±0.20	−0.13	−0.02	−0.03	0.009

**Table 3 cancers-14-02250-t003:** Overexpression of DNA replication genes and telomere maintenance genes in CRC compared to paired normal tissue stratified by tumor grade, tumor stage and MSI status.

Stratification	DNA Replication Genes		Telomere Maintenance Genes	
	Fold Change (95%CI)		Fold Change (95%CI)	
**Grade**				
Low-Grade	1.27	(1.25–1.28)	1.28	(1.25–1.31)
High-Grade	1.11	(1.09–1.14)	1.11	(1.06–1.16)
** *p* **	2.04 × 10^−26^		2.97 × 10^−8^	
**Stage**				
Stage-1	1.31	(1.28–1.34)	1.42	(1.35–1.49)
Stage-2	1.22	(1.20–1.25)	1.19	(1.15–1.24)
Stage-3	1.22	(1.20–1.23)	1.22	(1.18–1.25)
** *p* **	8.58 × 10^−7^		5.54 × 10^−8^	
**MSI status**				
MSS	1.23	(1.22–1.25)	1.25	(1.22–1.28)
MSI	1.23	(1.21–1.26)	1.22	(1.18–1.26)
** *p* **	0.801001		0.34	

## Data Availability

All supporting data are presented in the tables presented in the main manuscript and as additional Appendix A.

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
