# Peer review of "Relative Telomere Length Change in Colorectal Carcinoma and Its Association with Tumor Characteristics, Gene Expression and Microsatellite Instability"

_cancers, 2022, doi:10.3390/cancers14092250_

Round 1

Reviewer 1 Report

The manuscript " Relative Telomere Length Change in Colorectal Carcinoma and its Association with Tumor Characteristics, Gene Expression, and Microsatellite Instability" by Kibriya et al. study the RTL shortening in CRC tissue compared with healthy tissue from the same patient, and its relationship with different clinico-pathological parameters such as MSI, grade, location, or mutation status of KRAS and BRAF.

The study was conducted using fresh samples from 165 CRC patients from, Bangabandhu Sheikh Mujib Medical 80 University (BSMMU), Dhaka, Bangladesh.

The analysis for this set of samples revealed results that corroborate previous findings from other laboratories performed in different populations.

There are several points that need revision / correction,

  • In my opinion the paper will benefit if the Figure lettering was bigger.
  • There is a small typo on line 107 “The details of the assay is described “
  • The description of the analysis carried out is very extensive and detailed. I think that the reading of the article would improve if the writing of the results section was simplified.

Reviewer 2 Report

The present study investigated relative Telomere Length (RTL) in colorectal cancer (CRC) in relationship to matched normal tissue and characteristics of patients and tumors. This study revealed that RTL shortening in CRC was associated with up-regulation of DNA replication genes, Cyclin dependent kinase genes, and down-regulation of caspase gene reducing apoptosis. These results are interesting, as they unveil novel data on telomere in colorectal cancer. In the other hand, they are limited in scope.

The manuscript has been well written, and the data nicely interpreted. Several issues should, however, be addressed.

  • Characteristics of 71 CRC cases utilized for gene expression study should be reported.
  • The choice of the method for assessing RTL should be discussed; in particular, why assessment of RTL is not adjusted for age and gender. It should be justified why ratio TL was not used.
  • A figure shouldbe used to present different values of paired RTL
  • A result is cited in discussion paragraph and not reported in results paragraph (“We found WDR79 (WRAP53) gene to be overexpressed in CRC tumor with MSI”). This point needs to be clarified.
  • The following manuscripts need to be cited and discussed:
    • Suraweera N, et al. Relative telomere lengths in tumor and normal mucosa are related to disease progression and chromosome instability profiles in colorectal cancer. Oncotarget. 2016 Jun 14;7(24):36474-36488. doi: 10.18632/oncotarget.9015.
    • Nersisyan L, et al. Telomere Length Maintenance and Its Transcriptional Regulation in Lynch Syndrome and Sporadic Colorectal Carcinoma. Front Oncol. 2019 Nov 5;9:1172. doi: 10.3389/fonc.2019.01172. eCollection 2019.
    • Takagi S, et al. Relationship between microsatellite instability and telomere shortening in colorectal cancer. Dis Colon Rectum. 2000 Oct;43(10 Suppl):S12-7. doi: 10.1007/BF02237220.
